# Spinal Pain, Chronic Health Conditions and Health Behaviors: Data from the 2016–2018 National Health Interview Survey

**DOI:** 10.3390/ijerph20075369

**Published:** 2023-04-03

**Authors:** Katie de Luca, Patricia Tavares, Haiou Yang, Eric L. Hurwitz, Bart N. Green, Hannah Dale, Scott Haldeman

**Affiliations:** 1Discipline of Chiropractic, School of Health, Medical and Applied Science, CQ University, Brisbane, QLD 4701, Australia; 2Department of Clinical Education, Canadian Memorial Chiropractic College, Toronto, ON M2H 3J1, Canada; 3Center for Occupational and Environmental Health, University of California, Irvine, CA 92093, USA; 4Office of Public Health Studies, Thompson School of Social Work & Public Health, University of Hawaii, Mānoa, Honolulu, HI 96822, USA; 5Employer Based Integrated Primary Care Health Centers, Stanford Health Care, San Diego, CA 92121, USA; 6Department of Publications, National University of Health Sciences, Lombard, IL 60148, USA; 7Department of Neurology, University of California, Irvine, CA 92093, USA

**Keywords:** back pain, low back pain, neck pain, cardiovascular diseases, hypertension, diabetes mellitus, obesity, comorbidity, health behavior, spinal pain

## Abstract

Spinal pain and chronic health conditions are highly prevalent, burdensome, and costly conditions, both in the United States and globally. Using cross-sectional data from the 2016 through 2018 National Health Interview Survey (n = 26,926), we explored associations between spinal pain and chronic health conditions and investigated the influence that a set of confounders may have on the associations between spinal pain and chronic health conditions. Variance estimation method was used to compute weighted descriptive statistics and measures of associations with multinomial logistic regression models. All four chronic health conditions significantly increased the prevalence odds of spinal pain; cardiovascular conditions by 58%, hypertension by 40%, diabetes by 25% and obesity by 34%, controlling for all the confounders. For all chronic health conditions, tobacco use (45–50%), being insufficiently active (17–20%), sleep problems (180–184%), cognitive impairment (90–100%), and mental health conditions (68–80%) significantly increased the prevalence odds of spinal pain compared to cases without spinal pain. These findings provide evidence to support research on the prevention and treatment of non-musculoskeletal conditions with approaches of spinal pain management.

## 1. Introduction

Low back and neck pain (spinal pain) are considered the leading causes of years lived with disability in most countries and age groups [1]. In the 2021 Global Burden of Disease study, low back pain continues to be the greatest cause of disability burden worldwide, with 12.5% and 11.5% of years lived with disability due to low back pain attributed to modifiable risk factors smoking and elevated body mass index, respectively [2]. In adults aged 50 years and over, low back pain, ischemic heart disease and diabetes were all ranked in the top ten causes of disability adjusted life years globally [3], with ischemic heart disease and diabetes both in the top ten causes of death globally (1st and 10th, respectively). The costs of treating these diseases are very high. Spending during the year 2020 in the US was estimated at USD 134.5 billion for low back and neck pain, USD 111.2 billion for diabetes and USD 89.3 billion for ischemic heart disease.

Several studies have reported a relationship between spinal conditions and chronic health conditions such as cardiovascular disease [4,5,6], hypertension [7,8], diabetes [9,10], and obesity [11]. A population-based study conducted in the United States reported that adult Americans with spinal pain had an increased prevalence for comorbid conditions compared with those who had low back pain only or neck pain only [12]. In Australia, a number of chronic conditions (including cardiovascular conditions, diabetes and obesity) have been shown to be associated with spinal pain in older women with arthritis, with an incremental increase in the risk of spinal pain associated with increasing comorbidity count [13]. A scoping review conducted by the Global Spine Care Initiative reported a range of health conditions such as cardiovascular disease, hypertension, diabetes and obesity associated with major categories of spinal conditions [14].

Health behaviors have also been shown to be associated with spinal pain. In an early hallmark study on this topic, data from the 2002 US National Health Interview Survey (NHIS) revealed that current smokers, those with no leisure-time activity, and those with insomnia or trouble falling asleep were more likely to have spinal pain [12]. These findings were supported by data from the 2009–2012 NHIS, demonstrating significant associations between self-reported lower back pain and including leisure-time physical inactivity, current or former smoking, current or former alcohol drinking, and short sleep duration [15]. A biological gradient associated with exposure to smoking cigarettes and back pain was demonstrated with data from the 2012 NHIS, showing that the number of cigarettes smoked per day for current smokers was higher for those with back pain [16].

The 2016–2018 NHIS data releases have made it possible to combine years of survey data for questions related to physical and cognitive functioning and mental health conditions. This creates the opportunity to thoroughly analyze the associations of chronic health conditions, health behaviors, functional disability, and mental health conditions with spinal pain. Therefore, this study investigated the prevalence of four chronic health conditions in adults with spinal pain; explored associations between spinal pain and chronic health conditions; and investigated the influence that a set of confounders may have on the associations between spinal pain and chronic health conditions.

## 2. Materials and Methods

### 2.1. Data

The data used for this analysis are from the 2016 through 2018 NHIS. The NHIS is an annual cross-sectional health survey of the civilian noninstitutional population in the United States. Three data modules of the NHIS included in our analysis were Person, Sample Adult, and Functioning and Disability. The Person file included all members interviewed in the selected household and the Sample Adult data file included one adult sampled in the selected households. The Functioning and Disability file included about half of the respondents in the Sample Adult for 2016 and 2017. The Functioning and Disability questionnaire was administered to all respondents in the Sample Adult for 2018. The three data modules for each year were merged with matching household ID and Individual ID, and then the three years of the data were appended.

### 2.2. Measurements

#### 2.2.1. Spinal Pain

Four spinal pain variables included in the analysis were “low back pain only”, “neck pain only”, “spinal pain” and “no spinal pain”. These were determined through two question items: “Did you experience low back pain in the past three months?” and “Did you experience neck pain in the past three months?”. “Yes” to the question regarding low back pain and “No” to the question regarding neck pain defined “low back pain only”. “Yes” to the question regarding neck pain and “No” to the question regarding low back pain defined “neck pain only”. “Yes” to question regarding low back pain and neck pain defined “spinal pain”. “No” to both the questions regarding low back pain and neck pain defined “no spinal pain”.

#### 2.2.2. Chronic Health Conditions

We investigated four chronic health conditions: cardiovascular conditions, hypertension, diabetes and obesity. Cardiovascular conditions were assessed through a question that asked whether the respondent had been told by a doctor or other health professional that they had coronary heart disease, angina (or angina pectoris), heart attack (or myocardial infarction), stroke, or any other heart disease not already mentioned. Hypertension was assessed through a question that asked whether the respondent had been told by a doctor or other health professional on two or more different visits that they had hypertension or high blood pressure. Diabetes was assessed through a question that asked whether the respondent had been told by a doctor or other health professional that they had diabetes. BMI was calculated based on self-reported height and weight. Obesity was defined as a BMI ≥ 30 and severe obesity defined as a BMI ≥ 40.

#### 2.2.3. Confounders

Confounders included in the analysis were health behaviors, functional disability cognitive impairment, mental health conditions and demographic characteristics, which are described in detail below.

##### Health Behaviors

We investigated tobacco use, leisure-time physical activity, and sleep problems. Tobacco use was defined as current smokers assessed through 2 question items: smoking at least 100 cigarettes in their lifetime and still currently smoking (every day or some days). Leisure-time physical activity (with respect to aerobic activity reflects the federal “Physical Activity Guidelines for Americans, 2nd edition). This guideline recommended that for substantial health benefits, adults should perform at least 150 to 300 min a week of moderate intensity or 75 to 150 min a week of vigorous intensity aerobic physical activity, or an equivalent combination. In this study, leisure-time physical activity variables were categorized in the following way: “Physically inactive” denoted participating in no leisure-time aerobic activity that lasted at least 10 min; “Insufficiently active” denoted participating in aerobic activities for 10 min or more but less than 150 min per week; and “Sufficiently active” denoted participating in moderate intensity leisure-time physical activity 150 min or more per week, or participating in vigorous intensity leisure-time physical activity 75 min or more per week, or an equivalent combination. The reference group used in the analysis was “Sufficiently active”. Sleep problems were assessed with three questions on number of times having trouble falling asleep, staying asleep, and taking medication for sleep in the past week. Respondents were defined as having a sleep problem if they had >4 times in the past week trouble falling asleep, trouble staying asleep, or taking medication for sleep.

##### Functional Disability

Functional disability was assessed as difficulty performing 9 physical activities. In this set of 9 questions, respondents were asked about the degree of difficulty they experienced performing the following physical activities by themselves and without using any special equipment; walking a quarter of a mile (or three city blocks); standing for 2 h; stooping, bending, or kneeling; climbing 10 steps without resting; sitting for 2 h; reaching over one’s head; using one’s fingers to grasp or handle small objects; lifting or carrying a 10 pound object (such as a full bag of groceries); and pushing or pulling a large object (such as a living room chair). In addition, 3 other types of activities were assessed: going out for events, participating in social activities, and relaxing at home. The response categories consisted of “Not at all difficult”, “Only a little difficult”, “Somewhat difficult”, “Very difficult”, “Can’t do at all”, or “Do not do this activity”. Respondents were defined as having a functional disability if they responded, “Very difficult”, “Can’t do at all”, or “Do not do this activity”.

##### Cognitive Impairment

Cognitive impairment was assessed through the question “Do you have difficulty remembering or concentrating?” Respondents were defined as having cognitive impairment if they responded “a lot of difficulty” or “unable to” remember or concentrate.

##### Mental Health Conditions

We investigated the mental health conditions of anxiety and depression. Anxiety was assessed based on responses to the questions: “How often do you feel worried, nervous, or anxious?” and “Thinking about the last time you felt worried, nervous, or anxious, how would you describe the level of these feelings?”. For the frequency question, answer choices were “daily”, “weekly”, “monthly”, “a few times a year”, or “never”. For the severity question, answers were, “a little”, “a lot”, “somewhere in between a little and a lot”, “refused”, “not ascertained”, and “don’t know”. Respondents were defined as having anxiety, if they responded “daily” or “weekly” to the frequency question and “a lot” or “in between a little and a lot” to the intensity question. Depression was assessed based on responses to the questions: “How often do you feel depressed?” and “Thinking about the last time you felt depressed, how depressed did you feel?”. These 2 questions had the same response set as for anxiety. Respondents were defined as having depression if they responded “daily” or “weekly” to the frequency question and “a lot” or “in between a little and a lot” to the intensity question.

##### Demographic Characteristics

Demographic characteristics included four variables: sex, age, education and earnings. Age was coded into 7 age groups: 18–24 years, 25–34 years, 35–44 years, 45–54 years, 55–64 years, 65–74 years, and 75 years and above. The reference group used in the analysis was 18–24. Race and ethnicity were coded into five groups: Hispanic, Non-Hispanic White, Non-Hispanic Black, and Non-Hispanic Asian and Others. The Non-Hispanic White was used as the reference group. Socioeconomic status variables assessed included education and income (earnings). Education included 5 groups: less than high school, high school/general educational development, some college, Bachelor’s/Associate degree, and Master/Doctorate/professional degree. “Less than high school” was used as the reference group. Earnings were assessed through a question on total earnings last year, with 5 categories: <USD 14,999, USD 15,000–USD 24,999, USD 25,000–USD 44,999, USD 45,000–USD 74,999 and USD 75,000 and above. The lowest earnings group, <USD 14,999, was used as the reference group.

### 2.3. Statistical Analysis

To account for the complex sampling design that involves stratification and clustering of the NHIS, we used the variance estimation method in Stata 12, in [17], to compute weighted descriptive statistics and measures of associations. We divided the weight by three for the three combined years of data, 2016–2018. Descriptive analysis reports the characteristics of the study population, the prevalence of chronic health conditions, health behaviors, functional disability, cognitive impairment and mental health conditions by spinal pain status.

Measures of associations between spinal pain and chronic health conditions were conducted through three sets of multinomial logistic regression models. In these models, “no spinal pain” was used as the reference group. Each set of multinomial logistic regression models controlled for various groupings of confounders. Model 1 controlled for demographic variables (age, sex, and race/ethnicity) and socioeconomic variables (education and earnings). Model 2 controlled for health behaviors in addition to demographic and socioeconomic variables. Model 3 controlled for functional disability, cognitive impairment and mental health conditions, in addition to health behaviors, demographic and socioeconomic variables.

## 3. Results

After the three data files were appended for the 3 years, the sample was 55,220. After cases with missing values were excluded, the study sample was 26,926 (see Appendix A for data flow chart). Table 1 shows the prevalence of neck pain only, low back pain only, and spinal pain by age, sex, race/ethnicity, education, and earnings. These conditions were most common in the older age groups, with low back pain being the most prevalent condition at all ages. Spinal pain was more common in females and non-Hispanic Whites. The prevalence of neck pain only was highest in those with a Bachelor’s degree of education and above, while low back pain and spinal pain was more common in those with less than a college education. A similar pattern emerged with earnings, where the prevalence of neck pain only was highest among those with higher earnings, while those with lower earnings had a higher prevalence of low back pain only or spinal pain.

Figure 1 shows the prevalence of cardiovascular conditions, hypertension, diabetes, and obesity in the general population, by neck pain only, by low back pain only, and by spinal pain. All conditions were more prevalent in those reporting neck and low back pain (alone or together) than in the general population, with the exception of a slightly lower prevalence of obesity in those with neck pain only (26.9% vs. 30.5% in the general population).

Appendix A shows the prevalence of smoking, leisure-time physical activity, sleep problems, functional disability, cognitive impairment and mental health conditions in the general population, no spinal pain, neck pain only, low back pain only, and spinal pain. The prevalence of smoking, physical inactivity, and sleep problems was highest among those with spinal pain (24.5%, 35.9%, and 5.6%, respectively). The prevalence of functional disability, cognitive impairment and mental health conditions was highest among those with spinal pain (50.3%, 7.1% and 28.4%, respectively).

Table 2 shows Model 3 for the associations between cardiovascular conditions and spinal pain variables. Cardiovascular conditions increased the prevalence odds of spinal pain by 58%, of low back pain only by 15% and of neck pain only by 27% compared to cases without spinal pain and after adjusting for all confounders. For spinal pain, the health behaviors of tobacco use (46%), being insufficiently active (19%) and sleep problems (180%) significantly increased the prevalence odds compared with cases without spinal pain. For spinal pain, functional disability (324%), cognitive impairment (90%), anxiety (76%) and depression (70%) significantly increased the prevalence odds compared with cases without spinal pain.

Table 3 shows Model 3 for the associations between hypertension and spinal pain variables. Hypertension increased the prevalence odds of spinal pain by 40%, of low back pain only by 31% and of neck pain only by 23% compared to cases without spinal pain and after adjusting for all confounders. For spinal pain, the health behaviors of tobacco use (46%), being insufficiently active (17%) and sleep problems (181%) significantly increased the prevalence odds of spinal pain compared to cases without spinal pain. For spinal pain, functional disability (323%), cognitive impairment (94%), anxiety (768) and depression (71%) significantly increased the prevalence odds of spinal pain compared to cases without spinal pain.

Table 4 shows Model 3 for the associations between diabetes and spinal pain variables. Diabetes increased the prevalence odds of spinal pain by 25%, of low back pain only by 20% and of neck pain only by 12% compared to cases without spinal pain and after adjusting for all confounders. For spinal pain, the health behaviors of tobacco use (45%), being insufficiently active (20%) and sleep problems (183%) significantly increased the prevalence odds of spinal pain compared to cases without spinal pain. For spinal pain, functional disability (329%), cognitive impairment (100%), anxiety (77%) and depression (72%) significantly increased the prevalence odds of spinal pain compared with cases without spinal pain.

Table 5 shows Model 3 for the associations between obesity and spinal pain variables. Obesity had increased the prevalence odds of low back pain only by 34% and spinal pain by 17% compared to cases without low back pain only and without spinal pain and after adjusting for all confounders. For spinal pain, the health behaviors of tobacco use (50%), being insufficiently active (19%) and sleep problems (184%) significantly increased the prevalence odds of spinal pain compared to cases without spinal pain. For spinal pain, functional disability (331%), cognitive impairment (99%), anxiety (80%) and depression (68%) significantly increased the prevalence odds of spinal pain compared with cases without spinal pain.

Appendix A, shows the three models for the associations between each chronic health condition and spinal pain variables.

## 4. Discussion

From a large sample representative of the US population, we determined that chronic health conditions increased the prevalence odds of spinal pain by 58% for cardiovascular conditions, by 40% for hypertension, by 25% for diabetes and by 17% for obesity after controlling for confounders. The adjusted odds attenuate going from one model to the next but even after controlling for confounders, each chronic health condition is associated with higher odds of spinal pain than that of low back pain only and neck pain only. This study supports earlier conclusions that spinal pain is a critical health problem, [12,15], and reveals a multitude of physical and psychosocial comorbidities that are associated with spinal pain. Sadly, however, after 20 years of research, studies are still needed to address modifiable health behaviors and psychological risk factors shown here in people with low back, neck and spinal pain.

Individuals with chronic low back pain have been shown to experience more difficulty in managing their health, particularly in utilizing health information on optimizing lifestyle habits [18]. In this study, we found health behaviors such as tobacco use, being physically inactive or insufficiently active, and sleep problems significantly increased the prevalence odds of spinal pain. Sleep problems had the highest increases in prevalence odds ratios for spinal pain by 180–184%. In a recent systematic review of the association between sleep and chronic spinal pain, sleep was a strong predictor for the development of chronic spinal pain potentially due to poor sleep exacerbating responses to nociceptive stimuli and thus increasing existing pain hypersensitivity with variations in pain sensitivity occurring throughout the day [19]. Further, tobacco use also increased the prevalence odds for spinal pain by 45–50% and being insufficiently active increased the prevalence odds for spinal pain by 17–20%, when compared with patients without spinal pain.

Another finding of this study is that functional disability, cognitive impairment and mental health conditions were strongly associated with spinal pain. While functional disability had the highest increased prevalence odds of spinal pain (324–331%), cognitive impairment increased the prevalence odds of spinal pain by 90–100%. Regarding potential mechanisms underlying suboptimal cognitive performance in people with chronic low back pain, altered activity in the cortex and neural networks, grey matter atrophy, microglial activation and neuroinflammation, and comorbidities have been discussed in the literature [20]. In particular, depression and insomnia were identified as comorbid conditions that may accelerate cognitive decline in people with low back pain [20], both of which increased the prevalence odds for spinal pain, in all chronic health conditions in this study.

Strine and Hootman previously estimated the 3-month US prevalence of neck pain and low back pain at 9.3% [12], while in this study, we estimated the 3-month prevalence of spinal pain at 12.3%. The most recent Global Burden of Disease study estimates reveal a significant increase in the number of prevalent cases of low back pain globally since 1990 (60.4%), with projections suggesting that in 2050, there will be 843 million prevalent cases of low back pain globally [2]. The 2018 Lancet low back pain series highlighted the roles of advice and education that support self-management, physical, and psychological interventions, especially as first-line treatments for low back pain [21,22,23]. Also, a multidisciplinary approach to the assessment and treatment of spinal pain based on the associations found with comorbidities, lifestyle and behavioral factors has been suggested. Ramond-Roquin et al. [24] recently suggested taking into account comorbidity in patients with chronic low back pain and adopting a more comprehensive, patient-centered and coordinated approach. Such an approach may include screening for serious diseases, identifying barriers to improvement, managing motivations and expectations, embracing interprofessional collaboration and providing longitudinal continuity of care [24].

The NHIS is a population-based study with an annual response rate of approximately 70%. The NHIS allowed us to combine low back and neck pain data at the same point in time to explore spinal pain. Strengths of our study include using a large sample size (n = 26,926) and a national representative sample of the US population. Our study limitations include NHIS data being self-reported and that nondifferential misclassifications of spinal pain, chronic health conditions and confounders may underestimate true associations. The NHIS is a cross-sectional survey and it is not possible to draw conclusions about probable causal pathways between spinal pain, chronic health conditions, and confounders with this study design.

It is unclear why these variables have such strong associations. These associations could merely represent co-occurrence, defined as the presence of two or more health concerns [25] because all of the confounders are highly prevalent. Co-occurrence does not imply an underlying relationship, whether it be causal, complicating, or reciprocal [26]. Other health-related variables (such as psychosocial risk factors, diet, and workplace related risk factors) may influence these associations and therefore the results. Future longitudinal studies investigating alternative hypotheses represent fertile and important grounds to better understand the associations between spinal pain, comorbid conditions, health behaviors, functional disability, and mental health concerns.

An implication of this study would be for clinicians, including nurses, physicians, chiropractors, physical therapists, and psychologists, to consider the modifiable health factors documented in this study when counseling patients with spinal pain. Therefore, counseling for people with spinal pain and comorbidities may include improving leisure-time physical activity, improving sleep quality and reducing tobacco use.

## 5. Conclusions

Four chronic health conditions (cardiovascular conditions, hypertension, diabetes, and obesity) significantly increased the prevalence odds of spinal pain. For all chronic health conditions, tobacco use, being insufficiently active, sleep problems, cognitive impairment, and mental health conditions, significantly increased the prevalence odds of spinal pain compared to cases without spinal pain. These findings provide evidence to support research on the prevention and treatment of non-musculoskeletal conditions with approaches to spinal pain management.

## Figures and Tables

**Figure 1 ijerph-20-05369-f001:**
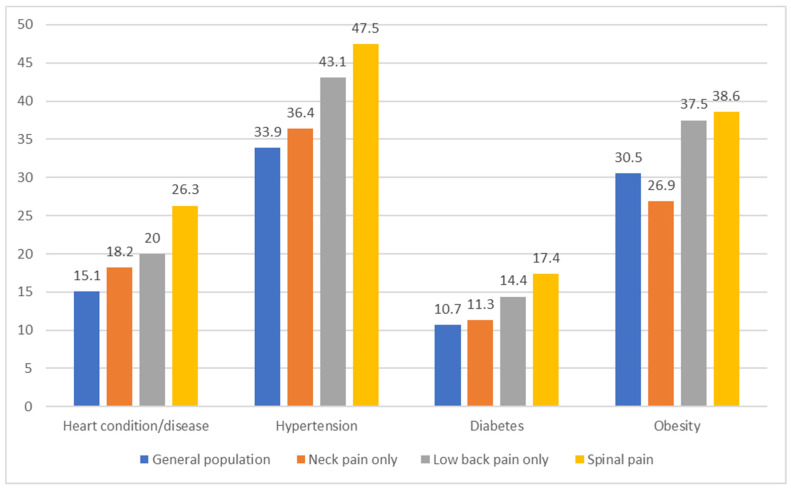
Prevalence of neck pain only, low back pain only and spinal pain in the US adult population, by chronic health conditions, 2016–2018.

**Table 1 ijerph-20-05369-t001:** Prevalence of low back pain only, neck pain only and spinal pain by demographic and socioeconomic variables in the US adult population, 2016–2018.

Demographic and Socioeconomic Variables	General Population	Neck Pain Only	Low Back Pain Only	Spinal Pain
Total, %		5.4	21.7	12.3
Age Group				
18–25	26.4	2.4	9.4	3.9
25–34	14.8	4.5	17.9	8.6
35–44	13.7	5.1	19.6	11.6
45–54	14.2	5.9	20.7	15.0
55–64	14.1	6.3	23.6	15.2
65–74	10.0	5.3	25.2	12.5
75 and above	6.80	5.6	26.4	11.3
Sex				
Female	51.1	5.8	21.8	13.3
Male	48.9	4.6	20.0	10.0
Race/Ethnicity				
Non-Hispanic White	61.8	5.8	22.1	12.5
Non-Hispanic Black	12.9	3.5	21.1	10.4
Hispanic	18.1	4.3	18.1	11.0
Non-Hispanic Asian and Others	7.2	4.6	15.6	8.2
Education				
Less than high school	27.6	4.6	24.7	14.6
High school/GED	21.3	4.7	23.9	12.7
Some college	15.5	5.1	21.9	12.8
Bachelor’s/Associate’s degree	25.9	5.5	18.9	10.7
Master’s/Doctorate/professional	9.7	6.1	16.9	8.3
Earnings				
<USD 14,999	18.5	4.9	11.1	19.5
USD 15,000–USD 24,999	13.6	4.8	10.5	20.8
USD 25,000–USD 44,999	25.0	4.5	10.5	19.1
USD 45,000–USD 74,999	22.6	5.9	8.7	18.4
USD 75,000 and above	20.2	5.5	8.0	17.4

**Table 2 ijerph-20-05369-t002:** The associations between cardiovascular conditions and spinal pain variables in the US adult population, 2016–2018 *.

Chronic Health Condition and Confounders	No Spinal Pain	Neck Pain Only			Low Back Pain Only			Spinal Pain		
		OR	95% CI	P	OR	95% CI	P	OR	95% CI	P
Cardiovascular conditions	Reference	1.27	(1.02,1.58)	0.03	1.15	(1.00,1.31)	0.05	1.58	(1.35,1.85)	<0.001
Health behaviors										
Current smokers		1.08	(0.90,1.30)	0.386	1.37	(1.25,1.51)	<0.001	1.46	(1.28,1.65)	<0.001
Leisure-time physical activity										
Sufficiently active		Reference			Reference			Reference		
Physically inactive		0.92	(0.76,1.12)	0.417	1.05	(0.95,1.17)	0.32	0.94	(0.82,1.08)	0.367
Insufficiently active		1.10	(0.92,1.31)	0.285	1.11	(1.01,1.21)	0.04	1.19	(1.06,1.33)	0.002
Sleep problems		1.81	(1.58,2.07)	<0.001	1.70	(1.57,1.84)	<0.001	2.80	(2.52,3.12)	<0.001
Functional disability		2.01	(1.62,2.48)	<0.001	2.61	(2.31,2.94)	<0.001	4.24	(3.67,4.89)	<0.001
Cognitive impairment		1.45	(0.77,2.73)	0.246	1.06	(0.71,1.59)	0.76	1.90	(1.35,2.67)	<0.001
Mental health conditions										
Anxiety		1.51	(1.22,1.87)	<0.001	1.39	(1.23,1.58)	<0.001	1.76	(1.53,2.03)	<0.001
Depression		1.36	(1.02,1.82)	0.039	1.24	(1.02,1.50)	0.03	1.70	(1.41,2.05)	<0.001

* Estimates adjusted for demographic and socioeconomic factors, health behaviors, functional disability, cognitive impairment and mental health conditions (Model 3).

**Table 3 ijerph-20-05369-t003:** The associations between hypertension and spinal pain variables in the US adult population, 2016–2018 *.

Chronic Health Condition and Confounders	No Spinal Pain	Neck Pain Only			Low Back Pain Only			Spinal Pain		
		OR	95% CI	P	OR	95% CI	P	OR	95% CI	P
Hypertension	Reference	1.23	(1.05,1.43)	0.01	1.31	(1.2,1.43)	<0.001	1.40	(1.25,1.56)	<0.001
Health behaviors										
Current smokers		1.09	(0.91,1.30)	0.344	1.38	(1.25,1.51)	<0.001	1.46	(1.29,1.65)	<0.001
Leisure-time physical activity										
Sufficiently active		Reference			Reference			Reference		
Physically inactive		0.91	(0.75,1.11)	0.35	1.04	(0.94,1.15)	0.455	0.93	(0.81,1.07)	0.329
Insufficiently active		1.09	(0.91,1.30)	0.346	1.09	(0.99,1.20)	0.071	1.17	(1.05,1.31)	0.005
Sleep problems		1.81	(1.58,2.07)	<0.001	1.69	(1.57,1.83)	<0.001	2.81	(2.53,3.13)	<0.001
Functional disability		2.00	(1.62,2.48)	<0.001	2.54	(2.26,2.87)	<0.001	4.23	(3.67,4.88)	<0.001
Cognitive impairment		1.45	(0.77,2.74)	0.249	1.06	(0.71,1.58)	0.784	1.94	(1.39,2.71)	<0.001
Mental health conditions										
Anxiety		1.51	(1.22,1.86)	<0.001	1.39	(1.23,1.57)	<0.001	1.78	(1.54,2.05)	<0.001
Depression		1.38	(1.03,1.84)	0.031	1.23	(1.02,1.49)	0.03	1.71	(1.42,2.06)	<0.001

* Estimates adjusted for demographic and socioeconomic factors, health behaviors, functional disability, cognitive impairment and mental health conditions (Model 3).

**Table 4 ijerph-20-05369-t004:** The associations between diabetes and spinal pain variables in the US adult population, 2016–2018 *.

Chronic Health Condition and Confounders	No Spinal Pain	Neck Pain Only			Low Back Pain Only			Spinal Pain		
		OR	95% CI	P	OR	95% CI	P	OR	95% CI	P
Diabetes	Reference	1.12	(0.88,1.44)	0.353	1.2	(1.05,1.38)	0.009	1.25	(1.05,1.49)	0.01
Health behaviors										
Current smokers		1.10	(0.92,1.32)	0.29	1.37	(1.24,1.51)	<0.001	1.45	(1.28,1.65)	<0.001
Leisure time physical activity										
Sufficiently active		Reference			Reference			Reference		
Physically inactive		0.9	(0.74,1.09)	0.283	1.03	(0.93,1.15)	0.537	0.95	(0.82,1.09)	0.424
Insufficiently active		1.08	(0.90,1.29)	0.428	1.11	(1.01,1.22)	0.029	1.20	(1.08,1.35)	0.001
Sleep problems		1.83	(1.59,2.10)	<0.001	1.69	(1.56,1.83)	<0.001	2.83	(2.54,3.16)	<0.001
Functional disability		2.15	(1.75,2.66)	<0.001	2.65	(2.35,2.99)	<0.001	4.29	(3.7,4.96)	<0.001
Cognitive impairment		1.17	(0.62,2.23)	0.63	1.03	(0.69,1.56)	0.869	2.00	(1.42,2.81)	<0.001
Mental health conditions										
Anxiety		1.47	(1.18,1.82)	<0.001	1.41	(1.25,1.59)	<0.001	1.77	(1.53,2.05)	<0.001
Depression		1.34	(0.99,1.81)	0.058	1.23	(1.02,1.48)	0.033	1.72	(1.43,2.08)	<0.001

* Estimates adjusted for demographic and socioeconomic factors, health behaviors, functional disability, cognitive impairment and mental health conditions (Model 3).

**Table 5 ijerph-20-05369-t005:** The associations between obesity and spinal pain variables in the US adult population, 2016–2018 *.

Chronic Health Condition and Confounders	No Spinal Pain	Neck Pain Only			Low Back Pain Only			Spinal Pain		
		OR	95% CI	P	OR	95% CI	P	OR	95% CI	P
Obesity	Reference	0.82	(0.71,0.96)	0.011	1.34	(1.25,1.45)	<0.001	1.17	(1.07,1.28)	0.001
Health behaviors										
Current smokers		1.06	(0.89,1.28)	0.504	1.41	(1.28,1.55)	<0.001	1.50	(1.32,1.7)	<0.001
Leisure-time physical activity										
Sufficiently active		Reference			Reference			Reference		
Physically inactive		0.95	(0.78,1.15)	0.57	1.04	(0.93,1.15)	0.502	0.95	(0.82,1.09)	0.435
Insufficiently active		1.13	(0.95,1.35)	0.178	1.07	(0.97,1.18)	0.163	1.19	(1.06,1.33)	0.003
Sleep problems		1.82	(1.58,2.08)	<0.001	1.70	(1.57,1.83)	<0.001	2.84	(2.54,3.16)	<0.001
Functional disability		2.14	(1.73,2.65)	<0.001	2.47	(2.19,2.79)	<0.001	4.31	(3.73,4.98)	<0.001
Cognitive impairment		1.46	(0.77,2.76)	0.246	1.05	(0.70,1.58)	0.804	1.99	(1.43,2.78)	<0.001
Mental health conditions										
Anxiety		1.48	(1.20,1.84)	<0.001	1.40	(1.24,1.59)	<0.001	1.80	(1.56,2.08)	<0.001
Depression		1.42	(1.06,1.91)	0.019	1.22	(1.01,1.48)	0.042	1.68	(1.39,2.03)	<0.001

* Estimates adjusted for demographic and socioeconomic factors, health behaviors, functional disability, cognitive impairment, and mental health conditions (Model 3).

## Data Availability

The datasets analyzed during this study are available in the National Center for Health Statistics repository, https://www.cdc.gov/nchs/nhis/data-questionnaires-documentation.htm (accessed on 18 January 2022).

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
