# Peer review of "Spinal Pain, Chronic Health Conditions and Health Behaviors: Data from the 2016–2018 National Health Interview Survey"

_ijerph, 2023, doi:10.3390/ijerph20075369_

Round 1
Reviewer 1 Report
Dear authors, we appreciate your effort and show intrest in your paper.
The paper ,, Spinal pain, chronic health conditions and health behaviors: an analysis of the 2016–2018 National Health Interview Survey”, takes in consideration the relation between back pain and chronic disease( mental and physical).
However, there are several mentions:
Table 1 shows the prevalence of cervical and lumbar pain related to demographic elements and the relation with the 4 chronic pathologies of your study. Table 2 - you don’t refere to gender, ethnicity, education and income. Same for tables 3,4,5.
It would have been more interesting to observe the relationship between spinal pain in sex subgroups and the other mentioned parameters, after reading lines 206-211,,Females and non-Hispanic Whites were more likely than males to report these conditions. Those with a college education were more likely to report neck pain only, while those with less than a college education more likely to report low back pain only or spinal pain. A similar pattern emerged with earnings, where prevalence of neck pain only was highest among those with higher earnings, while those with lower earnings more likely to report low back pain only or spinal pain.”
Lines 220-227 ,, Supplementary material 2 shows the prevalence of smoking, leisure time physical activity, sleep problems, functional disability, cognitive impairment, anxiety, and depression in the general population and by neck pain only, low back pain only, and spinal pain. Compared to the general population, the prevalence of smoking, not being sufficiently active, and sleep problems were all higher among those with neck pain only, low back pain only and spinal pain. Compared to the general population, the prevalence of physical and cognitive impairment and anxiety and depression were all higher among those with neck pain only, low back pain only and spinal pain.”
Here you present the prevalence of other pathologies (smoking, cognitive disability,..) in the study group compared to general population. In my opinion, it would have been a better choice to also point out the number of cases who experienced spinal pain in a specific region (cervical, lumbar), divided in subgroups by ethnicity, education, income.
Scheme 1 outlines an interesting fact, but further detailed research to determine the impact of ethnicity, education and income on the permanent caracter of the spinal pain.
Also cognitive impairment, anxiety and sleep disorders are more freqvent in Hispanic or non-Hispanic population?
Also would have been nice to put emphasis on the most affected gender and age-subgroups.
I appreciate you acknowledge the limitations of the study and its potential for further research.
My comments were only ment to improve your work. Best wishes!
Author Response
Reviewer # 1
- Table 1 shows the prevalence of cervical and lumbar pain related to demographic elements and the relation with the 4 chronic pathologies of your study. Table 2 - you don’t refere to gender, ethnicity, education and income. Same for tables 3,4,5.
- Response: Tables 2, 3, 4 and 5 are tables reporting the models for the associations between the 4 chronic pathologies and spinal pain variables. In each of the models, sex, ethnicity, education and earnings are reported as they are confounders. Please see the footnote of each table where these variables have been mentioned.
- It would have been more interesting to observe the relationship between spinal pain in sex subgroups and the other mentioned parameters, after reading lines 206-211, Females and non-Hispanic Whites were more likely than males to report these conditions. Those with a college education were more likely to report neck pain only, while those with less than a college education more likely to report low back pain only or spinal pain. A similar pattern emerged with earnings, where prevalence of neck pain only was highest among those with higher earnings, while those with lower earnings more likely to report low back pain only or spinal pain.”
- Response: Thank you for your interest in sex and ethnicity subgroups. We agree that this was interesting, but in the scope of the paper and to remain under the word count we discussed the findings of the inferential statistics rather than the descriptive statistics.
- Lines 220-227, Supplementary material 2 shows the prevalence of smoking, leisure time physical activity, sleep problems, functional disability, cognitive impairment, anxiety, and depression in the general population and by neck pain only, low back pain only, and spinal pain. Compared to the general population, the prevalence of smoking, not being sufficiently active, and sleep problems were all higher among those with neck pain only, low back pain only and spinal pain. Compared to the general population, the prevalence of physical and cognitive impairment and anxiety and depression were all higher among those with neck pain only, low back pain only and spinal pain.” Here you present the prevalence of other pathologies (smoking, cognitive disability) in the study group compared to general population. In my opinion, it would have been a better choice to also point out the number of cases who experienced spinal pain in a specific region (cervical, lumbar), divided in subgroups by ethnicity, education, income.
- Response: Again, thank you for this suggestion. Spinal pain variables were our key variables of interest, with chronic health conditions and health behaviors our secondary variables of interest, Therefore we do feel that presenting the health behaviors by each spinal pain variable is the correct presentation.
- Scheme 1 outlines an interesting fact, but further detailed research to determine the impact of ethnicity, education and income on the permanent caracter of the spinal pain. Also cognitive impairment, anxiety and sleep disorders are more freqvent in Hispanic or non-Hispanic population? Also would have been nice to put emphasis on the most affected gender and age-subgroups.
- Response: We do appreciate that the reviewer is interested in these sociodemographic variables, however it was not the objective of this study to explore these.
- I appreciate you acknowledge the limitations of the study and its potential for further research. My comments were only ment to improve your work. Best wishes!
- Response: Thank you for your comments, so helpful!
Reviewer 2 Report
Dear Authors, your paper is interesting, and well written, has many typos, and is statistically incomplete or flawed.
My unwelcome news is that I miss a statistical test method to illustrate your result. You say, “Variance estimation method was used to compute weighted descriptive statistics”
Otherwise said: I miss results in the Results chapter. I fail to see why you present three Models and only stipulate the Model 3 in your description of results. See below my remark 28.
Instead of three Models, which is strange if you do not use them, you should do one multinomial test, selecting the significant results.
Further details you find below:
Details, typo’s, suggestions for improvements and discussions
1. In the title you say ‘… an analysis of the 2016–2018 National Health Interview Survey.’ Shouldn’t this read‘ using the 2016–2018 National Health Interview Survey.’?
The reason for my word replacement is that you do not analyze the National Health Interview Survey. On the contrary: you gather data from the National Health Interview Survey, to draw conclusions;
2. In Line 7 you type Australia in capitals, while in lines 8, 12 you do not. Please make it consistent. I understand that the shortcut name USA is capitalized. Please complete the address at line 14 and put the symbol for the corresponding author after a name;
3. In over 13 places a blank space is missing: after a dot and/ or a comma. Example of such a missing blank is in: “ … groups.(1) …”, “… globally.(2) …”, “… et al.,(24) …”, etc;
4. Everywhere you should replace ‘pre-vention’ by ‘prevention’;
5. Everywhere you should replace ‘there-fore’ by ‘therefore’;
6. Everywhere you should replace ‘with-out’ by ‘without’;
7. Everywhere you should replace ‘in-to’ by ‘into’;
8. Line 123, ‘vigor-ous’ should read ‘vigorous’;
9. Line 287, ‘re-search’ should read ‘research’;
10. Line 362, ‘be-haviors’ should read ‘behaviors’;
11. In all cases your reference numbering misses squared [brackets]. This violates the style of the journal;
12. Table 1 is too much for me as a reader. Please put it in a polar chart, such as in https://www.mdpi.com/1660-4601/16/22/4446 ;
13. Lines 93-99, are unclear to me, it is said too repetitive. Please make sure that readers can understand it;
14. Line 119, please omit the double use of smoke: ‘currently smoking (smoke every day or …’ should read ‘currently smoking (every day or …’;
15. Line 120, ‘day’ should read ‘days’;
16. Line 125, ‘categorised’ should read ‘categorized’;
17. Line 134, ‘ taken’ ’ should read ‘taking’;
18. Line 149, ‘having functional’ should read ‘having a functional’;
19. Line 153, ‘having having’ should read ‘having’;
20. Line 154, ‘difficulty’ should read ‘difficulties’;
21. Line 163, ‘in between’ should read ‘in-between’;
22. Line 134, ‘Socioeconomic’ should read ‘The socioeconomic’;
23. Line 183, Your Statistical Analysis is incomplete/not correct.
Would it be an improvement if you introduce a test of independence of such test could output independence beyond your test of normality of the data.\;
24. Lines 188, 209, ‘prevalence’ should read ‘the prevalence’;
25. Line 211, ‘education more’ should read ‘education were more’;
26. Between Line 212 and 213, in the left column in the expression of the interval, is the – sign missing;
27. Lines 242ff, 254ff, 266ff, these three tables show erroneous display: last digits and brackets are put at a second line below the appropriate number, the word ‘reference’ occurs in the wrong column;
28. Lines 242ff, 254ff, 266ff, all these three tables are preceded by a description of the first line in Model 3, pertaining to saying the aOR(adjusted Odds Ratio) of the first data line in Model 3 and the last column of the aORs in Models 3. Are these the results in the results chapter? I fail to see why you present three models, and only stipulate the Model 3;
29. Lines 268 and 276, both have a bold head ‘Table’, in of these is erroneous;
30. Line 269, what happened to the neck pain? Why is it absent in your description?
31. Line 287, ‘Sadly however, and after’ should read ‘Sadly, however, after’;
32. Line 287, You say after 20 years of research. Is this Luca’s own research? What does the remark say about which team members? You have many papers published
33. Line 291, ‘utilising’ should read ‘utilizing’;
34. Line 292, ‘mising’ should read ‘missing’;
35. Line 293, ‘physical inactivy’ should read ‘physical inactive’;
36. Line 312, ‘both which increased’ should read ‘both of which increased’;
37. Line 313, ‘condtions’ should read ‘conditions’;
38. Line 325, ‘and adopting of a more comprehensive’ should read ‘and adopting a more comprehensive’;
39. Line 327, ‘screening for serious disease’ should read ‘screening for serious diseases’;
40. Line 338, ‘why these variables seem to have such high associations’ should read ‘why these variables have such high associations’;
41. Line 341, ‘com-plicating’ should read ‘complicating’;
42. Lines 349 and 350, ‘counselling’ should read ‘counseling’;
43. Line 358, to me it seems that your conclusions should be: to invest in prevention and treatment. A conclusion to do more research is a negative statement about your results;
44. Why are in [1] the appendix pages 802 – 815 of your prior work not mentioned? See: Eur Spine J 2018 Sep;27(Suppl 6):802-815. doi: 10.1007/s00586-017-5393; https://pubmed.ncbi.nlm.nih.gov/29282539/ ;
45. References [2], and [3] are incomplete, or clobber more than one paper or are just wrong. For instance nr. [2] needs pages and has no issue number;
46. Your reference [3] misses authors.
48. Reference [13] has KE de Luca as an author. Should it read: K de Luca?
Author Response
Reviewer # 2
- My unwelcome news is that I miss a statistical test method to illustrate your result. You say, “Variance estimation method was used to compute weighted descriptive statistics” Otherwise said: I miss results in the Results chapter. I fail to see why you present three Models and only stipulate the Model 3 in your description of results. See below my remark 28. Instead of three Models, which is strange if you do not use them, you should do one multinomial test, selecting the significant results.
- Response: In submitting this paper we removed the text on the descriptive results of the first two models for each chronic health condition, due to word count. We felt that readers could orientate themselves to each table to investigate the building of the first three models, including firstly sociodemographic factors, then health behaviors and finally functional disability, cognitive impairment, depression, and anxiety. As a response to this comment, we have removed the data lines from Tables 2 to 5 which describe to Model 1 and Model 2 for each of the chronic health conditions. Therefore, in the text and in the tables, we are only presenting data from Model 3. There have been several changes in the Tables 2 – 5 results section removing “3 models” and replacing with “Model 3”. As we allude to the attenuation going from one model to the next in the discussion, the 3 models for the associations between each chronic health conditions and spinal pain variables have been provided in Supplementary material
- In the title you say ‘… an analysis of the 2016–2018 National Health Interview Survey.’ Shouldn’t this read‘ using the 2016–2018 National Health Interview Survey.’? The reason for my word replacement is that you do not analyze the National Health Interview Survey. On the contrary: you gather data from the National Health Interview Survey, to draw conclusions.
- Response: Thank you for the suggestion, but we feel that we also did not “gather” the data for the NHIS survey. As a results we have included “ : Data from the … ”, to show that we used an existing dataset.
- In Line 7 you type Australia in capitals, while in lines 8, 12 you do not. P
- Response:
- Please complete the address at line 14
- Response: Corrected.
- In over 13 places a blank space is missing: after a dot and/ or a comma. Example of such a missing blank is in: “ … groups.(1) …”, “… globally.(2) …”, “… et al.,(24) …”,
- Response: Thank you, these have been corrected throughout.
- We have replaced the hyphenations throughout the manuscript, but we would like to express that these may happen during the formatting of the manuscript and may be out of our control during the submission process.
- We have corrected spelling mistakes (such as categorised) and grammars (such as day) throughout the manuscript.
- In all cases your reference numbering misses squared [brackets]. This violates the style of the journal.
- Response: These have been corrected.
- Table 1 is too much for me as a reader. Please put it in a polar chart, such as in https://www.mdpi.com/1660-4601/16/22/4446.
- Table 1 is a standardised descriptive table of the sample of our population. As we have multiple frequencies, the presentation of the Table as a Polar Chart would not be appropriate. We have not made any changes.
- Lines 93-99, are unclear to me, it is said too repetitive. Please make sure that readers can understand it
- Response: As these are variables taken from the dataset, we have not changed the text that describes each of the very important, spinal pain questions and categorisation of the variables. It is important we define here, in the methods, low back pain only, neck pain only, spinal pain and no spinal pain. We have not made any changes in this section. We have not made any changes.
- Line 154, ‘difficulty’ should read ‘difficulties’;
- Response: As this is a variable taken from the dataset, we have not changed difficulty to difficulties. We have not made any changes.
- Line 183, Your Statistical Analysis is incomplete/not correct.
Would it be an improvement if you introduce a test of independence of such test could output independence beyond your test of normality of the data.- Response: Exploring the comorbid associations between spinal pains and a set of health conditions is the goal of this study. We consider the multinomial logistic regression model is an appropriate way of statistical analysis for this study, as we are not only able to explore comorbid associations between individual health condition (cardiovascular conditions, hypertension, diabetes and obesity) and spinal pain, but also to differentiate the strength of comorbid associations between the health condition and various locations of spinal pain (neck pain only or low back pain only) as well as multiple coexisting of spinal pain (neck pain and low back pain). Moreover, the multinomial logistic regression models are also able to control for confounders, including health behaviors, functional disability cognitive impairment, mental health conditions, and demographic characteristics. A test of independence can demonstrate the comorbid association between individual types of health condition and individual type of spinal pain. However, a test of independence is not able to control for confounders. Moreover, a test of independence cannot determine the strength of comorbid associations between the health condition and various locations of spinal pain as well multiple location of spinal pain, compared with the multinomial logistic regression model. We actually conducted the test of independence but decided not to present the findings, due to the length limit of the paper, and also due to the reasons listed above. Therefore, no change is made in the paper.
- Lines 188, 209, ‘prevalence’ should read ‘the prevalence’
- Response: These have been corrected throughout the manuscript.
- Lines 268 and 276, both have a bold head ‘Table’, in of these is erroneous.
- Response: Thank you. Table 5 on line 268 has been corrected (un-bolded).
- Line 269, what happened to the neck pain? Why is it absent in your description?
- Response: Because the odds ratio was 0.82.
- Line 287, ‘Sadly however, and after’ should read ‘Sadly, however, after’ . Line 287, You say after 20 years of research. Is this Luca’s own research? What does the remark say about which team members? You have many papers published.
- Response: These two comments have been corrected. The manuscript now reads “Research studies … “
- Line 358, to me it seems that your conclusions should be: to invest in prevention and treatment. A conclusion to do more research is a negative statement about your results.
- Response: Thank for the suggestion that we greatly appreciated. We have reviewed the conclusions in the abstract to better highlight the impact of our current analysis. The change includes the sentence, “All four chronic health conditions significantly increased the prevalence odds of spinal pain; cardiovascular conditions by 58%, hypertension by 40%, diabetes by 25% and obesity by 34%, controlling for all the confounders. For all chronic health conditions, tobacco use (45-50%), being insufficiently active (17-20%) sleep problems (180-184%), cognitive impairment (90-100%), and mental health conditions (68-80%), significantly increased the prevalence odds of spinal pain, compared to those without spinal pain.”
- Why are in [1] the appendix pages 802 – 815 of your prior work not mentioned? See: Eur Spine J 2018 Sep;27(Suppl 6):802-815. doi: 10.1007/s00586-017-5393; https://pubmed.ncbi.nlm.nih.gov/29282539/
- Response: This paper has been cited as Reference [1].
- References [2], and [3] are incomplete, or clobber more than one paper or are just wrong. For instance nr. [2] needs pages and has no issue number; Your reference [3] misses authors.
- Response: Reference 2 is a pre-print of an upcoming paper from Lancet Rheumatology, therefore does not have an issue number as yet. Reference #3 is correct; the manuscript does not name individual authors.
- Reference [13] has KE de Luca as an author. Should it read: K de Luca?
- Response:
- Between Line 212 and 213, in the left column in the expression of the interval, is the – sign missing;
- Response: Tables had errors and inconsistencies in formatting due to the nature of the document formatting changes. We corresponded with Nattaporn Pinthong on 3rd March 2023 who confirmed we can submit the tables as separate files. Table 1 is an individual file, whereas Tables 2-5 are in one word file. This has now been done and there should not be any problems with formatting tables.
- Lines 242ff, 254ff, 266ff, these three tables show erroneous display: last digits and bracketsare put at a second line below the appropriate number, the word ‘reference’ occurs in the wrong
- Response: Thank you. Please see the above response, and “References” has been corrected in the formatting of the tables.
- Lines 242ff, 254ff, 266ff, all these three tables are preceded by a description of the first line in Model 3, pertaining to saying the aOR (adjusted Odds Ratio) of the first data line in Model 3 and the last column of the aORs in Models 3. Are these the results in the results chapter? I fail to see why you present three models, and only stipulate the Model 3.
- Response: We were limited by word count, to put all the text of the tables for each chronic condition. As a response to this comment, we have removed the data lines from Tables 2 to 5 which pertain to Model 1 and Model 2 for each of the chronic health conditions. Therefore, in the text and in the tables, we are only presenting data from Model 3. There have been several changes in the Tables 2 – 5 results section removing “3 models” and replacing with “Model 3”. As we allude to the attenuation going from one model to the next in the discussion, the 3 models for the associations between each chronic health conditions and spinal pain variables have been provided in Supplementary material 3.
Round 2
Reviewer 2 Report
-